# Methods Favoring Homology-Directed Repair Choice in Response to CRISPR/Cas9 Induced-Double Strand Breaks

**DOI:** 10.3390/ijms21186461

**Published:** 2020-09-04

**Authors:** Han Yang, Shuling Ren, Siyuan Yu, Haifeng Pan, Tingdong Li, Shengxiang Ge, Jun Zhang, Ningshao Xia

**Affiliations:** State Key Laboratory of Molecular Vaccinology and Molecular Diagnostics, National Institute of Diagnostics and Vaccine Development in Infectious Disease, Collaborative Innovation Centers of Biological Products, School of Public Health, Xiamen University, Xiamen 361102, China; yanghan@stu.xmu.edu.cn (H.Y.); shulingren@stu.xmu.edu.cn (S.R.); yusiyuan@stu.xmu.edu.cn (S.Y.); panhaifeng@stu.xmu.edu.cn (H.P.); zhangj@xmu.edu.cn (J.Z.); nsxia@xmu.edu.cn (N.X.)

**Keywords:** gene editing, CRISPR/Cas9, HDR, NHEJ, repair pathway choice

## Abstract

Precise gene editing is—or will soon be—in clinical use for several diseases, and more applications are under development. The programmable nuclease Cas9, directed by a single-guide RNA (sgRNA), can introduce double-strand breaks (DSBs) in target sites of genomic DNA, which constitutes the initial step of gene editing using this novel technology. In mammals, two pathways dominate the repair of the DSBs—nonhomologous end joining (NHEJ) and homology-directed repair (HDR)—and the outcome of gene editing mainly depends on the choice between these two repair pathways. Although HDR is attractive for its high fidelity, the choice of repair pathway is biased in a biological context. Mammalian cells preferentially employ NHEJ over HDR through several mechanisms: NHEJ is active throughout the cell cycle, whereas HDR is restricted to S/G2 phases; NHEJ is faster than HDR; and NHEJ suppresses the HDR process. This suggests that definitive control of outcome of the programmed DNA lesioning could be achieved through manipulating the choice of cellular repair pathway. In this review, we summarize the DSB repair pathways, the mechanisms involved in choice selection based on DNA resection, and make progress in the research investigating strategies that favor Cas9-mediated HDR based on the manipulation of repair pathway choice to increase the frequency of HDR in mammalian cells. The remaining problems in improving HDR efficiency are also discussed. This review should facilitate the development of CRISPR/Cas9 technology to achieve more precise gene editing.

## 1. Introduction

Precise genomic editing based on programming nucleases is opening up the possibility of achieving desired genome editing outcome in vitro and in vivo, and ensure its safe and rapid adoption for genome engineering applications across biology [1,2]. Although the discovery of zinc-finger nucleases (ZFNs) and transcription activator-like effector nucleases (TALENs) has greatly improved genome editing efficacy, using these systems for gene editing at new target sites in the genome requires re-designing or re-engineering a new set of proteins, which restricts their broad application [3]. In this respect, the clustered regularly interspaced short palindromic repeats (CRISPR)/CRISPR-associated protein (Cas) system has revolutionized genome editing technologies, mainly owing to its ease-of-use and lower engineering costs [1]. To date, the CRISPR/Cas system has been employed for gene editing in a diverse range of organisms [4], and is entering the realm of therapeutic application in humans [4,5].

CRISPR/Cas systems can be classified into six distinct types (I–VI) based on the assortment of known cas genes and properties of the CRISPR ribonucleoprotein (crRNP) effector complexes [6,7]. The type II SpCas9 is the most widely used for genome engineering purposes [8]. SpCas9 is a protein that complexes with a guide RNA—either as a separate CRISPR RNA (crRNA) and an additional noncoding RNA-transactivating crRNA (tracrRNA) component or as a chimeric single-guide RNA (sgRNA) [9]—yielding an RNP that can bind and cleave target DNA [10]. DNA recognition by SpCas9 relies on a 20-nucleotide-long spacer and a protospacer adjacent motif (PAM; 5′-NGG) in a sequence-specific manner [11]. Following DNA binding through standard Watson–Crick base pairing, SpCas9 cleaves each strand via two distinct nuclease domains (HNH: target strand, RuvC: nontarget strand), inducing a blunt-ended DNA double-strand break (DSB) [12].

Introduction of a DSB at a specified genomic site is only the first step in gene editing. These lesions must be resolved by cellular DNA repair pathways, during which genomic DNA might be modified [13]. In general, mammals have evolved at least four DNA repair pathways, including two major pathways—nonhomologous end joining (NHEJ)/canonical NHEJ (c-NEHJ) and homology-directed repair (HDR)—and two additional pathways, namely, alternative NHEJ (alt-NHEJ) and single-strand annealing (SSA). The outcome of gene editing depends on the choice of repair pathway, especially between those involved in HDR and NHEJ (Figure 1).

HDR utilizes the sister chromatid as a template for repair and creates a desired and precise gene editing product. Conversely, NHEJ seals DSB with little or no homology, resulting in random gene editing products (insertions or deletions [indels] at the cleavage site) [14]. HDR allows for the use of an exogenous DNA template to generate almost any desired DNA change, and may therefore have a wider application range, especially for safer clinical use [3,15]. However, the frequency of HDR in nature is extremely low, and mammalian cells preferentially employ NHEJ over HDR through several mechanisms: NHEJ is active throughout the cell cycle except in mitosis, whereas HDR is restricted to S and G2 phases; NHEJ is faster than HDR; and NHEJ represses HDR through a series of mechanisms [16]. Therefore, strategies that manipulate the repair choice and favor HDR could assist in the utilization of CRISPR/Cas9 systems to achieve more precise genome editing.

## 2. The Mechanism of DSB Repair Used by the CRISPR/Cas9 System

Although HDR and NHEJ employ specific repair factors, both pathways use the same signaling propagation cascade to drive the cellular responses to the Cas9-induced DSB. Within seconds of the appearance of a DSB, three PI3K-like kinases (PIKKs) are activated, including ataxia telangiectasia mutated (ATM), ataxia telangiectasia and rad3-related (ATR), and DNA-dependent protein kinase (DNA-PK) [17]. H2AX is phosphorylated by a PIKK, yielding γH2AX, which spreads throughout the area surrounding the breakage site [18]. Subsequently, PIKK-phosphorylated mediator of DNA damage checkpoint protein 1 (MDC1) binds directly to γH2AX and recruits the E3 ubiquitin-protein ligases RING finger 8 (RNF8) and RING finger 168 (RNF168); the recruitment of RNF168 requires the RNF8-mediated formation of K63-linked ubiquitin chains [19]. RNF168 potently ubiquitylates H2A-type histones at K13 and K15 (H2AK13/15), generating recruitment platforms for a range of repair factors, including TP53-binding protein 1 (53BP1) and breast cancer 1 (BRCA1), which facilitate either NHEJ or HDR, respectively [20].

### 2.1. HDR

HDR occurs largely during the S/G2 phase, when an undamaged sister chromatid (or donor DNA) is available. The major step committing a DSB to HDR is 5′-to-3′ resection of the DNA end to form a 3′ single-stranded DNA overhang [21]. This process is initiated by the MRN (MRE11-RAD50-NBS1) complex, which also serves as a scaffold for amplifying the ATM signaling response to the DSB [22]. MRN recruits the C-terminal-binding protein interacting protein (CtIP) and initiates the resection process, whereby the NBS1 subunit generates short single-stranded tails [23]. Then, the exonuclease 1 (Exo1) and the DNA replication ATP-dependent helicase/nuclease DNA2 (Dna2)/ bloom syndrome protein (BLM) complex perform long-range DNA resection, resulting in a 3′ ssDNA tail [24,25]. Because free single-stranded DNA ends are very unstable, this 3′ ssDNA overhang is rapidly bound and shielded by replication protein A (RPA). With the assistance of recombination ‘mediators’, such as BRCA1, BRCA2, and partner and localizer of BRCA2 (PALB2), RPA is replaced by DNA repair protein RAD51 homolog 1 (RAD51), which then forms extended nucleoprotein filaments on the ssDNA [26,27]. The 3′ protein filament mediates homology searches and strand invasion of the homologous DNA template by the ssDNA, generating a displacement loop (D-loop) [28]. At least three subpathways are proposed to act after formation of the D-loop intermediate, depending on whether one or two Holliday junctions are formed [29]. Finally, the resolution junction is processed by a group of enzymes called resolvases that terminate the repair process and restore the chromosome to its undamaged state [22,30]. The sequence information from the sister chromatid or foreign DNA that restores the integrity of genomic DNA is the starting point for HDR-based precision editing (Figure 2).

### 2.2. NHEJ/c-NHEJ

NHEJ (or c-NHEJ) is constitutively active in all stages of the cell cycle and is the default DNA repair pathway in mammalian cells [31]. NHEJ is initiated by the binding of Ku70-Ku80 (KU) to blunt or near-blunt DNA ends [32]. Once KU is bound to the DSB ends, it then serves as a scaffold to recruit other c-NHEJ-related factors to the damage site. This step is important to protect the DNA from end resection. DNA-PKcs, the catalytic subunit of DNA-PK, is first recruited by KU, forming an active DNA-PKcs+KU complex [33]. DNA-PK assembly stimulates the kinase activity of DNA-PK and orchestrates c-NHEJ. DNA-PKcs phosphorylates an array of substrates, including artemis, X-ray repair cross complementing protein 4 (XRCC4), DNA ligase IV, and XRCC4-like factor (XLF) [34], which promotes the synapsis of DNA ends and facilitates the recruitment of end-processing and ligation enzymes. Aligned, compatible DNA termini can be directly ligated by the c-NHEJ factors, in case DNA termini were incompatible, one of the main exonucleases, artemis, cooperates with polymerase lambda (λ) and mu (μ) to prepare blunt DNA ends for the next stage. In the final step, the XRCC4-DNA ligase IV-XLF complex performs the ligation across gaps separately and independently of each other [35,36]. DNA bases are randomly added and removed through the activity of DNA polymerases and nucleases, resulting in small indels relative to the original genomic template, which constitutes the basis of NHEJ-based error-prone editing (Figure 2).

### 2.3. Two Additional Pathways

In addition to HDR and NHEJ, mammals have at least two additional DSB repair pathways: alt-NHEJ, defined as a c-NHEJ backup, and SSA. Unlike HDR, these two additional pathways are as highly error-prone as NHEJ.

Alt-NHEJ shares features of both NHEJ and HDR and repairs DNA DSBs by annealing, using 2–20 nt microhomology (MH) (also termed MH-mediated end-joining, MHEJ) [37,38]. Although the initial DNA resection through alt-NHEJ resembles that mediated by HDR, the MRN complex is recruited by poly [ADP-ribose] polymerase 1 (PARP1) to the broken sites. Notably, KU, which possesses an extremely high affinity for DNA ends, efficiently competes with PARP1 at the DSB and inhibits the alt-NHEJ pathway [39]. As an obligatorily DNA ligase IV-independent pathway, the ligation step in alt-NHEJ relies on DNA ligase 1/3, and may also involve XRCC1 [40,41] (Figure 1). Therefore, alt-NHEJ repairs a DSB by annealing small homologous regions on each side of the break, allowing for the precise control of deletion outcomes without the need for a DNA donor [42]. Several studies have recently reported relatively high frequencies of alt-NHEJ-based target integration using CRISPR/Cas9 [43,44]. However, alt-NHEJ shares some of the technical disadvantages associated with NHEJ: suffering from the indel problems at 5′ and 3′ junctions of DNA insertion sites [45].

SSA is typically initiated when DSBs occur at genomic loci where extensive homology (DNA repeats) exists between sequences at either side of the DSB; its homology information is derived from a complementary strand and not a homologous chromosome, as occurs with HDR [46]. Notably, instead of the RAD51-mediated synapsis used in HDR, in SSA, RAD52 aligns the RPA-coated, single-stranded DNA after resection. However, RAD51 displaces RPA and RAD52 from ssDNA, suggesting that HDR is favored over SSA. Moreover, deletion of RAD52 does not lead to significant DNA repair or recombination impairment in mammals [47]. In general, SSA is a RAD51-independent repair pathway that is enhanced when RAD51 is suppressed [48] (Figure 1). Although the SSA process utilizes homologous repeats to bridge DSB ends, SSA is regarded as a mutagenic process that can generate DNA deletions of up to several hundred base pairs between the DNA repeats.

## 3. The Mechanism Regulating the Choice between the HDR and NHEJ Repair Pathways 

Mammalian cells have evolved elaborate regulatory mechanisms to influence the choice of repair pathway, particularly that between NHEJ and HDR. By virtue of its roles in the initiation of the HDR process and simultaneous inhibition of NHEJ, DNA end resection is the critical regulator of DSB repair pathway choice [49].

### 3.1. The Regulation of DNA Resection by Cyclin-Dependent Kinases

The activity of cyclin-dependent kinases (CDKs) begins from the S phase of the cell cycle and continues through the G2 and M phases [50], providing stimulatory signals to the resection machinery. CDKs activate DNA resection mainly through the phosphorylation of multiple substrates, one of the most crucial of which is CtIP. In the S phase, the phosphorylation of CtIP at Thr847 by CDKs stimulates MRE11 endonuclease activity, which promotes HDR selection in the S phase [23,51]. Meanwhile, the CDK1-mediated phosphorylation of CtIP at Ser327 facilitates its binding to BRCA1, which helps BRCA1 counteract 53BP1 [52]. In addition to transmitting the initiation signal of DNA resection to CtIP, CDK activity also promotes long-range resection through the phosphorylation of Exo1 [53], the phosphorylation of Exo1 facilitates its recruitment to DSB via interactions with BRCA1. Apart from the direct boost to the relative factors involved in DNA resection, CDKs also induce DNA-end resection by weakening the impediments to this process. In the G1 phase, DNA helicase B (HELB) can be recruited to DSBs via interaction with RPA, and uses its 5′-to-3′ ssDNA translocase activity to inhibit the extensive resection activity of Exo1 and BLM-DNA2. The activity of CDK2 begins with S-phase progression, resulting in the export of HELB from the nucleus, its isolation from the DSB, and the consequent upregulation of DNA resection [54]. Additionally, CDKs can regulate the activity of ATP-dependent DNA helicase Q4 (RECQL4) to coordinate NHEJ or HDR. In the G1 phase, when overall CDK activity is low, RECQL4 interacts with Ku70 to promote NHEJ. In the S/G2 phase, CDK1 and CDK2 phosphorylate and activate RECQL4, inducing MRE11-RECQL4 interaction and recruitment of RECQL4 to DSBs [55,56] (Figure 3). 

### 3.2. The Regulation of DNA Resection by Ubiquitin

Ubiquitylation is an essential, reversible posttranslational modification that involves covalent attachment of ubiquitin to a substrate [57]. Ubiquitylation not only generates recruitment platforms for the coordinated assembly of various ubiquitin-binding domain (UBD)-containing repair factors to DSB sites [19], but also has important roles in determining which pathway is used to repair DSBs through regulation of DNA end resection [58].

Owing to its abundance and strong affinity for DNA, KU can associate with DNA ends within 5 s of damage in any phase of the cell cycle. At the late S-phase, DNA ends must be free of KU for initiation of DNA resection, and consequently HDR. However, MRN alone cannot displace KU from DSB ends [21]. KU release from DNA ends is known to depend on the cooperation between multiple ubiquitylation events. First, the short ssDNA overhang produced by MRE11 must be recognized by the zinc fingers of RNF138 [59]. KU is then actively displaced from the DSB, which is mediated by RNF138-dependent ubiquitylation of its Ku80 subunit. Interestingly, the E3 ligase RNF138 stimulates DSB end resection via a dual involvement. RNF138, in complex with members of the UBE2D family of E2 conjugating enzymes, also interacts with CtIP to foster its ubiquitylation and accumulation at DSBs, which further facilitates DNA resection [59]. The second mechanism that mediates KU dissociation is neddylation-dependent ubiquitylation; however, neddylation may only play a minimal role in KU release [60]. Regulation of DSB end resection via KU ubiquitylation may be more complex, and the other E3s, including RNF82 [61] and SCFFBXL1, also promote Ku80 ubiquitylation and removal from DSBs, with RNF8 being specifically required for this process in the G1 phase of the cell cycle.

Several E3 ubiquitin ligases have been found to contribute to the proteasome-dependent degradation of CtIP, thereby inhibiting DNA resection [62]. The anaphase-promoting complex/cyclostome (APC/C), an E3 ubiquitin ligase involved in cell cycle regulation, has also been demonstrated to specifically target CtIP. APC/C-mediated proteasomal degradation maintains the low expression of CtIP after mitotic exit as well as after DNA damage in the G2 phase, allowing efficient NHEJ and restricting HDR in G1 [63]. A similar mechanism has been demonstrated to operate in PIN-related CtIP proteasomal degradation; here, the binding of peptidyl-prolyl cis-trans isomerase NIMA-interacting 1 (PIN1) to phosphorylated CtIP, which is mediated by CDKs, results in PIN1-mediated isomerization of CtIP and leads to CtIP ubiquitylation and subsequent degradation [64]. Additionally, Kelch-like protein 15 (KLHL15), a substrate-specific adaptor for cullin-3 (CUL3)-based E3 ubiquitin ligases, catalyzes CtIP polyubiquitination and promotes its degradation.

### 3.3. The Battle between 53BP1 and BRCA1 for DNA Resection

53BP1 and BRCA1 are pivotal regulators of DSB repair choice between NHEJ and HDR [65]. 53BP1 was shown to negatively regulate resection in the G1 phase of the cell cycle, consequently inhibiting HDR; in contrast, BRCA1 promotes the removal of 53BP1 in the S phase to allow resection [66]. The mechanisms that allow 53BP1 to be replaced with BRCA1, and why it is active at this specific time, have not been fully elaborated yet.

Following DSB formation and activation of ATM signaling, phosphorylated (S/T-Q) 53BP1 is recruited to chromatin around DSBs through concomitant binding to histone H4 demethylated on lysine 20 (H4K20Me2) and K15-ubiquitylated histone H2A (H2AK15Ub) via its methyl-binding tandem Tudor domain and a ubiquitylation-dependent recruitment motif (UDR), respectively [67]. In the G1 phase, 53BP1, together with its key effectors RAP1-interacting factor 1 (RIF1) and PAX transactivation activation domain-interacting protein (PTIP), antagonizes BRCA1 activity. Together with REV7, the 53BP1-RIF1 complex strongly inhibits the recruitment of BRCA1 [68]. Recently, RIF1 was reported to recruit the newly identified Shieldin complex [69,70] and cell cycle regulator of NHEJ (CYREN) [71] to protect DSB ends from resection. Another 53BP1 partner, PTIP, recruits artemis, a well-known DSB end-processing nuclease involved in NHEJ, which limits the ability of RPA or RAD51 to bind DSB and initiate DNA resection [72]. 

Cells weaken the blockage of 53BP1 activity in DNA resection in a BRCA1-dependent manner [73]. At the S/G2 phase of the cell cycle, CDKs phosphorylate CtIP, inducing the formation of a CtIP-BRCA1 complex. The formation of this complex prevents the association between 53BP1 and RIF1, and possibly also the association between 53BP1 and PTIP, thereby directing repair toward the HDR pathway. However, various studies have suggested that BRCA1-CtIP interaction is not essential for resection per se, instead modulating the speed of this process [74]. Meanwhile, BRCA1 recruits CDK-phosphorylated UHRF1, a E3 ubiquitin ligase, to DSBs. UHRF1 then polyubiquitylates RIF1, leading to the dissociation of RIF1 from 53BP1 [75]. BRCA1 also weakens the barrier to 53BP1-dependent resection in a cell cycle- and ubiquitination-dependent manner. In the S phase, BRCA1 dimerizes with BRCA1-associated RING domain protein 1 (BARD1) to yield a functional E3 ligase. The BRCA1/BARD1 ligase modifies H2A at chromatin, thereby promoting the accumulation and activity of the chromatin remodeler SMARCAD1 (SWI/SNF-related matrix-associated actin-dependent regulator of chromatin), which then mobilizes 53BP1 and its effector proteins, thereby allowing the completion of resection [76]. Another approach mediated by BRCA1 involves the PP4C-dependent dephosphorylation of 53BP1, which facilitates RIF1 release [77] (Figure 4).

## 4. Methods to Enhance CRISPR/Cas9-Mediated HDR Choice

With greater knowledge of how the HDR repair pathway is suppressed in the G1 phase and stimulated in the S/G2 phases of the cell cycle, approaches can be developed that selectively disrupt error-prone repair pathways, particularly NHEJ, thereby facilitating precise HDR; directly promote HDR repair over NHEJ; or supply a favorable cellular context for HDR (S/G2 phase, pairing Cas9-induced DSBs with a DNA template).

### 4.1. Regulation of the Trade-Off between c-NHEJ and HDR

#### 4.1.1. Suppression of Key NHEJ Factors

Although cells that suffer DSBs in the S and G2 phases first attempt HDR to resolve the insults, c-NHEJ and HDR occur concurrently. This suggests that inhibiting the expression or function of factors essential for NHEJ may result in the channeling of DSB repair to the HDR pathway [78]. DNA ligase IV, which is responsible for sealing the break during the final step of c-NHEJ, may be a potential target for the inhibition of NHEJ [31]. The frequency of ZFN-mediated HDR is enhanced (70%) in flies with deficient DNA ligase IV relative to the wild type, suggesting that this may be a promising approach to follow in CRISPR/Cas9-related research [74,79]. SCR7 is a potent NHEJ inhibitor targeting on DNA ligase IV, and was initially harnessed by Srivastava et al. as a therapeutic strategy for cancer treatment in a mouse model. In 2005, three groups successfully employed SCR7 to inhibit NHEJ, thereby increasing CRISPR/Cas9-mediated HDR frequency. Maruyama et al. found that SCR7 treatment improved gene insertion efficiency by between 2- and 19-fold with inserts of differing lengths in mammalian cells. The authors also observed that co-injection of CRISPR-Cas9 components and SCR7 into zygotes improved the efficiency of HDR-mediated insertional mutagenesis, resulting in the generation of mice with insertions at multiple loci [80]. Embryonic stem cells do preferably use HDR, thus, herein the role of SCR7 as an HDR promoter might be questionable. Two other independent studies [81,82] also employed SCR7 to improve the frequency of precise gene modifications. SCR7 was identified as a NHEJ inhibitor and widely utilized in subsequent studies [83,84,85,86]. However, whether SCR7 really promotes HDR remains controversial, with several groups reporting that SCR7 exerts negligible effects on HDR activity [87,88,89,90,91,92]. Additional work is necessary to confirm the effects of SCR7 on HDR efficiency [93]. In addition to small molecule inhibitors, blockage of DNA ligase IV activity using other approaches, such as shRNA-mediated gene silencing and proteolytic degradation through E1B55K- and E4orf6-mediated ubiquitination of DNA ligase IV, can also increase the efficiency of HDR in mammalian cells [82].

Other attempts to suppress NHEJ have focused on targeting critical factors upstream of the DNA ligase IV pathway. KU is the first protein to bind to DSBs in c-NHEJ regardless of cell cycle stage, and cells will first attempt to repair DSBs by c-NHEJ if the ends are compatible. Owing to the role of KU in promoting NHEJ and suppressing HDR, Li et al. used KU-specific siRNA to downregulate the expression of Ku70 and Ku80, which resulted in a significant increase in the frequency of CRISPR/Cas9-mediated HDR in pig fibroblasts [94]. Further upstream of NHEJ repair, treating DNA-PK with two chemical inhibitors, NU7441 and KU-0060648, led to a twofold increase in the rate of HDR at DSBs induced by Cas9 in HEK-293T cells [95]. NU7441 was also successfully applied in zebrafish embryos, resulting in a marked enhancement of the HDR frequency (13.4-fold) [96]. Riesenberg and colleagues tested the effects of a series of potential small-molecule NHEJ inhibitors/HDR enhancers not previously used in Cas9-mediated gene editing studies and identified two novel and effective DNA-PKcs inhibitors, NU7026 and M3814 [97,98]. 

Given the importance of NHEJ for genome maintenance, inhibiting this repair pathway may lead to the accumulation of unrepaired DSBs in the cell, eventually leading to cell death [78] or late embryonic lethality [99]. Therefore, the safety of using these inhibitors in vivo would need to be carefully evaluated.

#### 4.1.2. In Favor of HDR Factors

Alternatively, HDR agonists may be considered a clinically safer option. RAD51, which is involved in strand exchange and the search for homology, plays a central role during HDR. The small molecule RS-1, identified through the screening of a 10,000 compound library, can enhance the binding activity of RAD51 [100]. Pinder et al. were the first to show that treatment with RS-1 increased Cas9-stimulated HDR by three- to sixfold in HEK-293A cells in a manner that was dependent on the locus and transfection method [101]. Subsequently, Song et al., applied RS-1 in rabbit embryos and achieved a modest improvement (two- to sixfold) in the knock-in rate. This agonist was also successfully employed for CRISPR/Cas9-mediated knock-in in zebrafish embryos [83] and human pluripotent stem cells (hPSCs) [90]. Recently, Kurihara et al. showed that RAD51 overexpression in neural progenitor cells enhanced knock-in efficiency by approximately twofold [102]. Although NHEJ is predominant in mammalian cells, HDR is more common in yeast [103]. Importantly, yeast RAD52 (yRAD52) can promote strand invasion of RPA-coated ssDNA in the presence of RAD51 (biological context), although its human analogue, hRAD52, cannot, suggesting that yRAD52 may be a candidate factor for enhancing HDR [104,105]. One promising study used this knowledge to show that overexpression of yRAD52 or a yRAD52-Cas9 fusion protein increased the frequency of HDR at different loci in both HEK-293T and porcine PK15 cells, irrespective of the template type (plasmid, PCR product, ssDNA) [106]. Application of the same fusion protein (yRAD52-Cas9) also resulted in a threefold increase in HDR in chicken cells using single-stranded donor oligonucleotides (ssODNs) as donor DNA [107]. Although several studies have reported that hRAD52 may not to be effective in promoting HDR (even promoting negative regulation) [104], Paulsen et al., found that ectopic expression of hRAD52 improved CRISPR-mediated HDR only if a ssODN template was used, presumably by promoting the engagement of the SSA pathway-based precise editing [108]. The strategy of fusion expression was also applied with CtIP, the key protein in DNA resection. A recent study demonstrated that, compared with Cas9 alone, HDR mediated by Cas9 fused with the N-terminal fragment of CtIP (including the multimerization domain and key CDK phosphorylation sites) was increased by twofold or more. This increase was achieved through the recruitment of endogenous CtIP at the cleavage site in human cell lines, iPS cells, and rat zygotes, although the Cas9-CtIP effect varied according to the locus [109]. Consistent with these findings, Tran et al., found that the HDR-related factor CtIP (full length), RAD52, and MRE11 could all increase the precise-editing efficiency by up to twofold in human cells when fused to Cas9 [110]. A novel strategy has been reported that uses chimeric Cas9 constructs, in which SpCas9 is fused to a 126-amino acid intrinsically disordered domain from the HSV-1 alkaline nuclease (UL12), which is not an HDR-relevant factor. In this approach, UL12 could increase the recruitment of the MRN complex to DSBs, thereby facilitating HDR [111].

Several HDR enhancers with yet unknown mechanisms of action on DSB repair have also been identified. Based on a reporter screen system, Yu et al., identified two small molecules, L755505 and Brefeldin, screened from among roughly 4000 small molecules, both of which enhanced HDR efficiency by approximately two- to threefold for knock-in of large fragments, and also led to a nine fold increase in the generation of point mutations [112]. However, treatment with L755505 induced only a slight HDR enhancement in porcine fetuses and had no effect on knock-in efficiency in induced pluripotent stem cells (iPSCs) [86,97].

#### 4.1.3. Manipulation of the Relationship between 53BP1 and BRCA1

As described above, 53BP1 is the critical regulator of the repair choice between NHEJ and HDR. Deletion of 53BP1 induces an increase in HDR [113], suggesting that inhibiting 53BP1 may be a promising tool to manipulate repair choice. Recently, Canny et al., screened an engineered ubiquitin variant library and discovered a genetically encoded inhibitor of 53BP1, inhibitor 53 (i53), that targets its tandem Tudor domain. This inhibitor acts by blocking the interaction between 53BP1 and H4K20Me2 and suppressing the accumulation of 53BP1 at DSBs, which stimulates Cas9-mediated HDR by up to 5.6-fold in different human and mouse cells [91]. Paulsen et al., suppressed 53BP1 function through transient expression of a dominant-negative murine form of 53BP1 (mdn53BP1). Ectopic expression of mdn53BP1 competitively antagonized 53BP1 recruitment to DSBs and improved Cas9-mediated HDR activity in HEK-293T cells, and this stimulatory effect was greatest when synergizing with the temporary expression of RAD52 [108]. Conversely, overexpression of dn53BP1 was reported not to be efficient in human cells, and the global suppression of NHEJ induced by untethered dn53BP1 was particularly toxic to human cells [114]. Alternately, these authors identified a small-sized, dominant-negative form of human 53BP1 (DN1S), and fusion of DN1S to Cas9 restricted the suppressive effect of dn53P1 around the DSB, avoiding the unwanted effects of global NHEJ inhibition. Recently, Nambiar and colleagues systematically examined the impact of DNA damage response (DDR) factors on Cas9-induced HDR, and identified RAD18 as a potent HDR enhancer (twofold). RAD18 binds H2AK15Ub with greater affinity than 53BP1, thereby inhibiting 53BP1 recruitment to DSBs [115] (Figure 5). Regarding 53BP1 antagonists, Pinder et al. showed that cells overexpressing wild-type BRCA1 or its hyper-recombination mutant variants (BRCA1K1702M, BRCA1M1775R) also exhibited increased rates of Cas9-induced HDR [101].

### 4.2. Cas9 Activity Paired with the HDR-Active Cell Cycle Phase

Considering that the activity of the HDR pathway is restricted to the late S and G2 phases of the cell cycle, restricting Cas9-induced DSB formation to S/G2 would likely improve HDR efficiency. Synchronizing cells to the HDR-permissive cell cycle phase could be achieved through pharmacological means in vitro, such as exposure to nocodazole, ABT751, or RO-3306. Combining nocodazole with direct delivery of preassembled Cas9-RNP complexes achieved controlled appearance of DSBs at G2 [116]. Moreover, Cas9-mediated HDR was increased by up to 1.38-fold in nocodazole-treated HEK-293T cells relative to unsynchronized cells (also effective in human primary neonatal fibroblast and embryonic stem cells) [116]. ABT751 exhibits the same biological function as nocodazole, and promoted a six-fold increase in targeting efficiency in iPSCs [117]. Lomova et al., reported that the rate of HDR increased in RO-3306 (selective CDK1 inhibitor)-treated human hematopoietic stem cells [118]. Recently, XL413, a novel small molecule targeting CDC7, which plays a key role in DNA replication initiation, was found to increase the efficiency of HDR by up to 3.5-fold in many contexts, including primary T cells, through a transient arrest in S-phase of cell cycle [119].However, these approaches may have limitations for in vivo use owing to potential toxicity. 

Timely expression of Cas9 could be an additional approach to synchronize DSB creation with HDR-permissive phases. Gutschner et al., developed a strategy to control the expression of hSpCas9 in the S and G2 phases of the cell cycle that was based on intracellular regulatory circuits. The APC/CDH1 complex is active in the late M and G1 phases and promotes the ubiquitination of a variety of proteins, including geminin. Fusing the first 110 amino acids of geminin with hSpCas9 (Cas9-geminin) resulted in its proteolytic destruction in the late M and G1 phases. This engineered Cas9-geminin construct increased the HDR rate by 1.87-fold compared with wild-type Cas9 in HEK-293T cells [88]. The efficacy of the hCas9-geminin approach was subsequently confirmed in human pluripotent stem cells and porcine fibroblasts [120,121], making it the most promising tool for enhancing HDR (Figure 6). Lomova et al., found that there was an additive effect between manipulation of Cas9 expression and cell cycle synchronization via RO-3306, which improved the HDR/NHEJ ratio by an average of 20-fold compared with the control condition in hPSCs [118]. In addition, manipulating the activity of CDKs can also increase HDR rates. Ye et al., reported that when active Cas9/sgRNA induced DSBs, Cas9/dgRNAi (inhibition) and Cas9/dgRNAa (activation) simultaneously inhibited Ku80 expression and stimulated CDK1, respectively, yielding an approximately fourfold improvement in HDR in several cell lines [122].

### 4.3. All the Components in One HDR Complex

It has been proposed that HDR efficiency can be significantly increased if donor molecules are readily available at the time of DNA cleavage [123]. Various attempts have been made to enhance HDR based on this hypothesis. Carlson-Stevermer et al. first introduced the all-in-one HDR complex, comprising Cas9-RNP and donor DNA, which increased the proximity of the CRISPR components and donor DNA inside cells. In their system, termed S1mplex, a modified sgRNA containing a streptavidin-binding aptamer was linked to a biotinylated ssDNA donor, generating the RNP-donor DNA complex. The S1mplex system stimulated an 18-fold increase in HDR when compared with unlinked components [124]. Different from the S1mplex system that requires extra sgRNA modification, Ma, M et al. devised a Cas9 variant that was fused to avidin via a flexible linker and bridged with biotin-modified ssDNA. This system, which was based on the affinity between avidin and biotin, achieved an HDR frequency of ~20% in mouse embryos [125]. Similarly, Savic et al., generated an RNP-ssDNA complex using the SNAP-tag, and obtained a 24-fold increase in HDR efficiency [126]. However, these three methods all require modifying the donor DNA or sgRNA, and thus the costs of these methods are high. Recently, Aird and colleagues proposed a new strategy to enhance HDR rates by covalently tethering ssDNA to the Cas9-RNP complex via a fused HUH endonuclease, which bypassed the need to modify the sgRNA or donor ssDNA [92] (Figure 7). For the “all in one” methods described above, the donor types are all single-stranded oligodeoxynucleotides (ssODNs), owing to the high frequency (~30%) of HDR they can support in mammalian cells and the ease of preparation [127,128]. However, the difficulty in generating long ssDNA (>200 nucleotides) has restricted the adoption of this method for the replacement of large fragments. Alternatively, dsDNA templates with long-length homology (1–2 kb, such as circular and linearized plasmids and PCR products) are usually used for generating large sequence changes. The type of donor dsDNA also influences the efficiency of HDR. Song et al. compared the performance of circular and linearized plasmid DNA donors using a modified traffic light reporter (TLR-3) system, and showed that linearized plasmids are two- to fourfold more effective than circular plasmids in inducing HDR in HEK cells. This could be explained by the fact that circular DNA would undergo nonspecific cleavage in the homology arm [129].

## 5. Conclusions and Perspectives

CRISPR/Cas9-induced HDR represents a tool to modify genomic DNA in a precise and controlled manner. However, owing to the preference of mammalian cells for NHEJ over HDR, the efficiency of HDR is significantly lower than that of NHEJ (25% vs. 75%). To date, numerous approaches to selectively disrupt NHEJ, directly boost the HDR repair pathway, or supply a favorable cellular context for overcoming the choice limitations of HDR using the CRISPR/Cas9 system have proven successful. Nevertheless, the enhanced application of these strategies depends on variable biological and experimental characteristics, including the type of cell and organism, genomic location, and experimental design (Table 1). Moreover, these methods face many safety issues. NHEJ is critical for genome stability, and plays an important role in the repair of DSB. Suppression of NHEJ-relevant factors or the choice regulator 53BP1 may lead to increased toxicity, both in vitro and in vivo. Cell synchronization using chemicals may also have limitations when used in vivo due to potential toxicity. In contrast, methods that favor HDR factors, timely delivery of Cas9, and all-in-one strategies have greater applicability. Combining various approaches to enhance HDR frequencies will likely be the optimal choice for the future. Finally, an increased understanding of the repair pathways would enable the identification of approaches that facilitate Cas9-based HDR in various contexts. For example, a small open reading frame (ORF) cell-cycle-specific inhibitor of c-NHEJ (CYREN) was recently identified, and overexpression or activation of CYREN is a potential strategy to enhance the frequency of Cas9-mediated HDR.

## Figures and Tables

**Figure 1 ijms-21-06461-f001:**
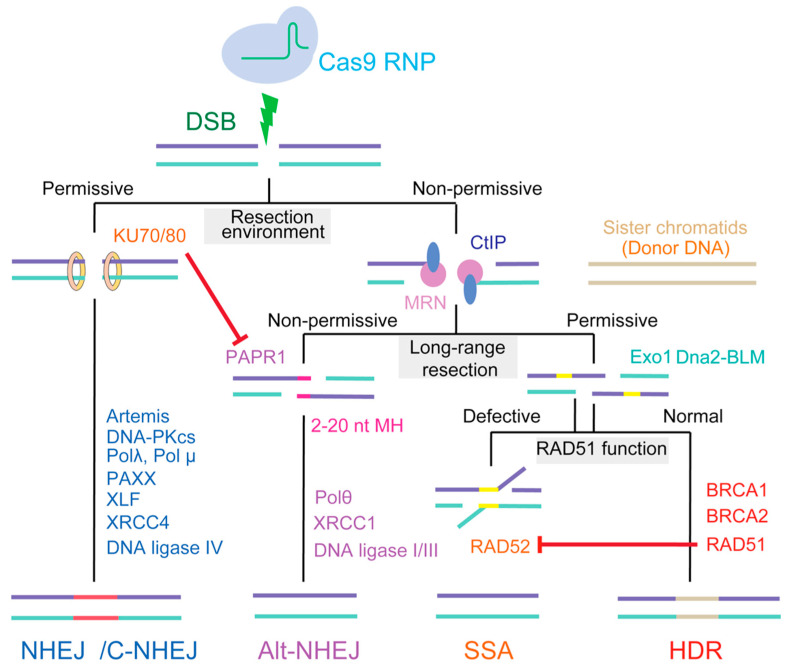
The hierarchy of repair pathways choice in mammalian cell. NHEJ (nonhomologous end joining) is the first choice DSB (double strands break) repair pathway in mammalian cells, occurring in all cell cycle phase. Preliminary end resection is necessary for the DSB repair according to HDR (homology-directed repair), SSA (single-strand annealing), and alt-NHEJ (alternative NHEJ) mechanisms. Resection during alt-NHEJ is relatively short, and extended resection favors HDR. SSA can function opportunistically on ssDNA (single strand DNA) ends or on recombination intermediates, especially RAD51, by hijacking the HDR process.

**Figure 2 ijms-21-06461-f002:**
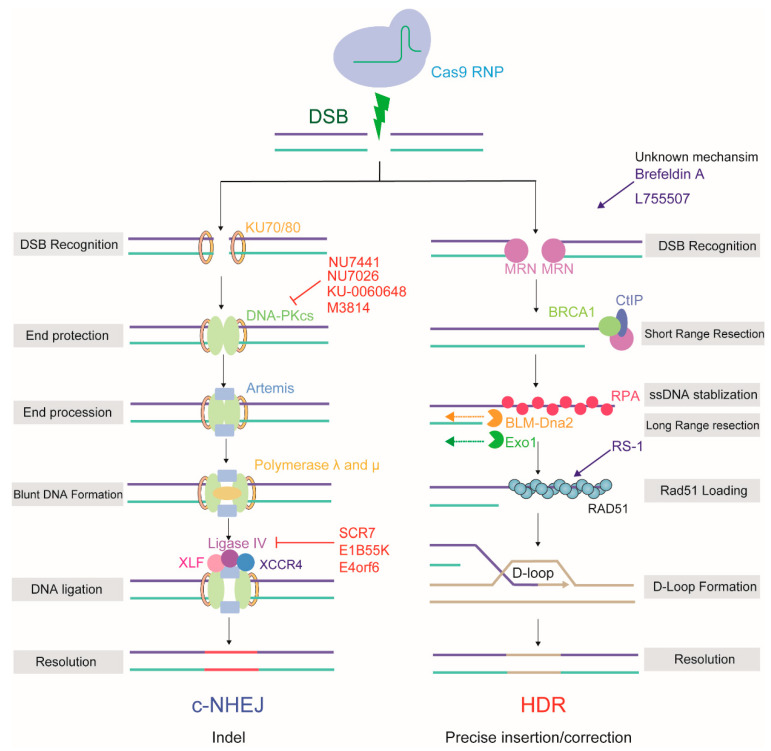
The mechanism and outcome of two major repair pathways: NHEJ and HDR, used by CRISPR/Cas9 induced-DSB in mammalian cells. NHEJ inhibitors are labeled in red, HDR activators are labeled in blue.NU7441, KU-0060648 and NU7026 suppress the DNA-PKcs; SCR7, E1B55K and E4orf6 suppress the ligase IV; RS-1 boosts RAD51; Brefeldin and AL755507 acts with unknown mechanism.

**Figure 3 ijms-21-06461-f003:**
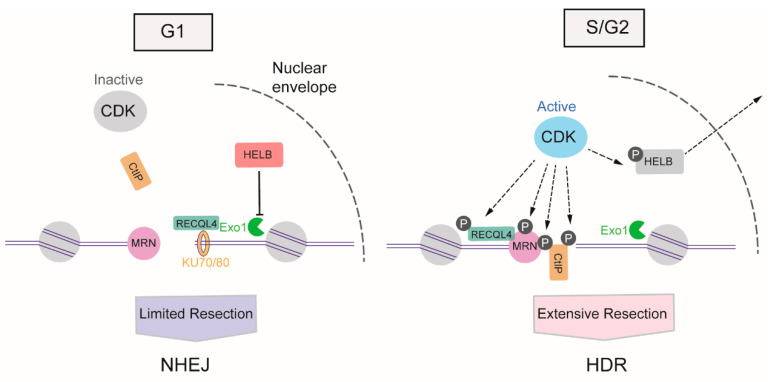
DNA resection-dependent regulation of DSB repair pathway choice by CDKs (cyclin-dependent kinases). CDKs activities arise from S phase and progresses in G2 and M phases, promoting the DNA resection through phosphorylation of multiple substrates, such as CtIP (C-terminal-binding protein interacting protein), MRN (MRE11-RAD50-NBS1)), HELB (helicase B), RECQL4 (ATP-dependent DNA helicase Q4) and Exo1 (exonuclease 1).

**Figure 4 ijms-21-06461-f004:**
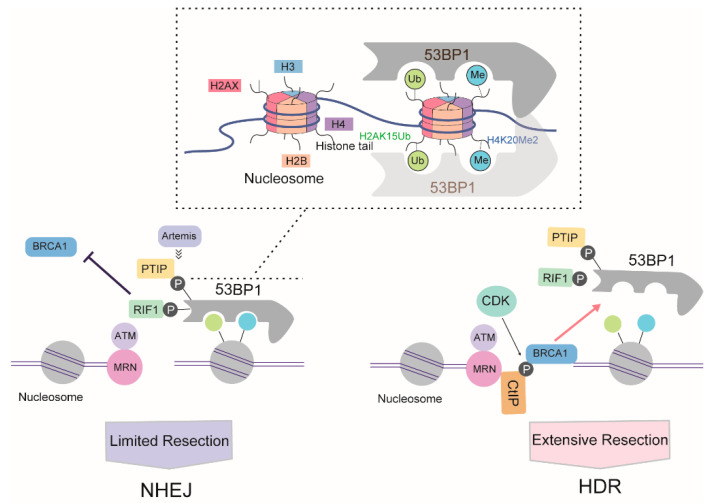
Proposed antagonistic relationship of 53BP1 and BRCA1 during DSB repair pathway choice based on DNA resection. The box above indicated the model of 53BP1 interaction with nucleosome.

**Figure 5 ijms-21-06461-f005:**
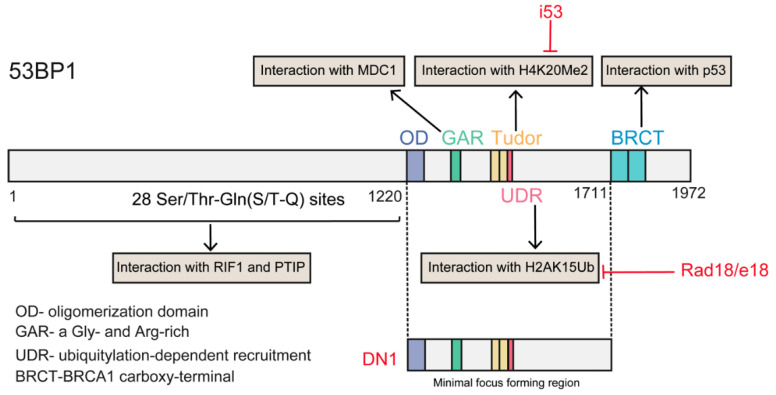
Structural domains of the 53BP1 protein and its interaction partners. The red color text indicated the different approaches of inhibiting 53BP1 function. The i53 targets in Tudor domain and interferes the interaction of 53BP1 with H4K20Me; Rad18/e18 interferes the interaction of 53BP1 with H2AK15Ub; DN1 a dominant negative version of minimal focus forming region.

**Figure 6 ijms-21-06461-f006:**
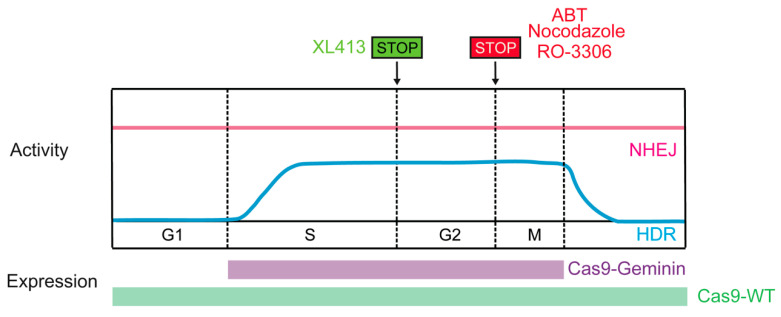
Methods restricting the Cas9-induced DSB to HDR-active cell cycle phases.

**Figure 7 ijms-21-06461-f007:**
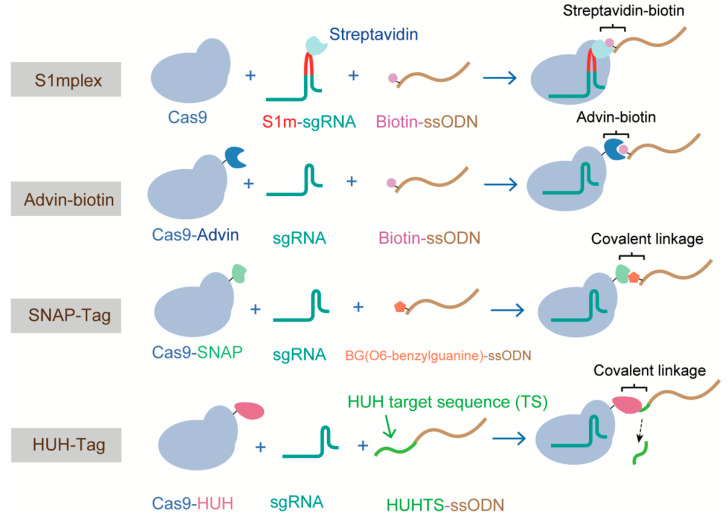
Methods tethering a ssODN donor template to Cas9 RNP, forming the all components in one HDR complex.

**Table 1 ijms-21-06461-t001:** Selected small molecules used in favoring the choice of Cas9 mediated-HDR.

Target	Treatment	HDR Increasement	Cell Type	Gene Locus	Editing Type	Ref.
DNA Ligase IV inhibitor	SCR7	3-fold	A549	TSG101	Stop cassette KI	[80]
		19-fold	Mel-JuSo	TSG101	Stop cassette KI	[80]
		13-fold(4.58–58.3%)	DC2.4	Tap1	Venus KI fluorescent gene KI	[80]
		4.5-fold(5–22.7%)	Mouse zygotes	Igk	LPETG KI	[80]
		5-fold(5–25%)	HEK293	AAVS1-TLR	GC	[82]
		10-fold(5.8–56.2%)	Mouse embryos	Tex1	GC	[81]
		2.5-fold(2.7–6.6%)	MCF-7	AAVS	GC	[85]
		10-fold(1.1–4.3%)	HCT-116	AAVS	MCS KI	[85]
		4-fold (5.6–11.2%)	Porcine fetal fibroblasts	eGFP	MCS KI	[86]
		1.9-fold (26.22–49.66%)	Primary porcine cell	ROSA26	Neomycin KI	[86]
		3.5-fold (1.9–6.6%)	MCF-7 /GFP-mut	-	GC	[86]
		1.7-fold (8.4–14.6%)	HCT-116 *Δ*TCT	β-catenin	GC	[86]
		3.7-fold (15–55%)	Zebrafish embryos	Ybx 1	S82A GC	[83]
		2.2-fold (18–39%)	Rat zygotes	Fabp2	Cre KI	[84]
		1.4-fold (46–64%)	Rat zygotes	Dbndd	Cre ER T2 KI	[84]
		NS	Rabbit embryos	RLL	EGFP KI	[87]
		NS	HEK-293T	rOSA26	Neomycin KI resistance gene KI	[88]
		NS	Porcine foetal Fibroblasts	Rosa26	EGFP KI	[89]
		NS	HSPC	Rosa26	EGFP KI	[90]
		NS	U2OS	LMNA-GFP	GC	[91]
		NS	HEK-293T	GAPDH	HIBIT KI	[92]
PK-cs inhibitor	NU7441	3~9-fold	MEF	TP53	GC	[95]
		2-fold (1.9–3.8%)	HEK293-TLR	-	GC	[95]
		13.5-fold (4–53%)	Zebra embryos	-	mCherry KI	[96]
		2.5-fold (8.6–21.5%)	A549	CD45	GFP KI	[90]
		2.4-fold (8–19.2%)	CD34+ HSPC	CD45 gene/GFP insertion	GFP KI	[90]
		1.3-fold (4.6–6%)	U2OS	LMNA	mClover KI insertion	[91]
	KU-0060648	2~8–fold	MEF	TP53	GC	[95]
		2.2-fold (1.9–4.1%)	HEK293-TLR	-	GC	[95]
	NU7026	1.6-fold (3.7–6.0%)	iPSCs	AAVS	KI	[97]
		1.7-fold (19–32%)	hiPSCs-409B2	FRMD7	GC	[98]
		2.4-fold (2.5–6%)	HEK293	GFP	KI	[130]
	M3814	4-fold (18–81%)	K562 cells-409B2 hiPSCs	FRMD7	GC	[98]
Rad51 agonist	RS-1	EP 3-fold(2.5–7.5%)epEP/electroporation	HEK293A	LMNA	Clover KI	[101]
		Lipo 6-fold(3.5–21%)/lipofection	HEK293A	LMNA	Clover KI	[101]
		8.5-fold (0.8–6.8%)	Rabbit embryos	RLL	EGFP KI	[87]
		2.1-fold (8–16.8%)	CD34+ HSPC	CD45 gene/GFP insertion	GFP KI	[90]
		1.6-fold(15–24%)	Zebrafish embryos	Ybx1	S82A GC	[83]
		NS	iPSC	CTNB1/PRDM14	mNeoGreen KI	[97]
Cell cycle Synchronization	Nocodazole	2.2-fold(9–20%)	HEK293T	EMX1, DYRK1	MCS KI	[116]
		1.7-fold (13–22%)	iPSC	PRDM14	mCherry KI	[131]
		1.7-fold (7–12%)	iPSC	CTNNB	mCherry KI	[131]
		1.4-fold (14–19.2%)	HEK293T	MALAT1	MCS KI	[88]
		6.7-fold (7.86–52.49%)	DiPSC	NEUROD	Puromycin KI	[117]
		3.4-fold (17.46–59.77%)	H1	NEUROD	Puromycin KI KI	[117]
	ABT	6.8-fold (7.86–53.50%)	DiPSC	NEUROD	Puromycin KI	[117]
		4.6-fold (17.46–79.91%)	H1	NEUROD	Puromycin KI	[117]
	RO-3306	1.3-fold(18.1–23.5%)	PBSC	β-globin	GC	[118]
Golgi apparatus	Brefeldin A	2-fold (17.7%–27.2%)	E14 mouse ESCs	Nanog	GFP KI	[112]
inhibitor		1.3-fold (13%–17%)	iPSC	CTNB1	KI	[131]
β3-adrenergic receptor agonist	L755507	3-fold (17.7–33.3%)	E14 mouse ESCs	Nanog	GFP KI	[112]
		8.9-fold (0.35–3.13%)	iPSC	SOD1	A4V GC	[112]
		2-fold (0.8–1.6%)	HUVEC	ACTA2	Venus KI	[112]
		1.9-fold (5.6–10.9%)	Porcine fetal fibroblasts	eGFP	GC	[86]
		NS	iPSC	CTNB1/PRDM14	mNeoGreen KI	[97]

NS: no significant; GC: gene correction; KI: knock in; *Δ*TCT: heterozygous Ser45 deletion; MCS: multiple cloning sites; -: random locus insertion; TLR: traffic light reporter.

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
