# Peer review of "Methods Favoring Homology-Directed Repair Choice in Response to CRISPR/Cas9 Induced-Double Strand Breaks"

_ijms, 2020, doi:10.3390/ijms21186461_

Round 1

Reviewer 1 Report

The authors reviewed the pathway of homology-directed repair (HDR) and how to control it in response to CRISPR-directed double-strand break. Since genome editing technology is simple in principle of operation, but the process and output are often complicated, researchers need to be careful about how to use it. In this review manuscript, authors described HDR process and key players in comparison with other cellular processes including NHEJ and SSA. The manuscript is well-described in terms of the issues associating with both HDR and Cas9. I have some minor comments as follows:

  1. line 361-364; the word "dn53P1" should be corrected to "dn53BP1".
  2. line 90;  "S/G1 phase" should be corrected to "S/G2 phase".
  3. Supplementary table 1 should be placed in the main text if possible, because of its importance.

Author Response

Dear Editors and Reviewers:

Thanks for your professional handling and consideration of our manuscript entitled “Methods Favoring Homology-Directed Repair Choice in Response to CRISPR/Cas9 Induced-Double Strand Breaks” (ijms-900814). The comments and advice from the reviewers are very helpful for revising and improving our manuscript. According to the reviewers’ comments and your advice, we revised relevant parts in the manuscript and tracked changes on the modified version of ijms-900814.In addition, the language has been edited by Charlesworth Author Services before submission. A file with line-to-line response to Reviewers’ comments was attached. I wish that the revised manuscript is now suitable to be published in “International Journal of Molecular Sciences”.

Thanks, and best wishes.

Yours sincerely,

Shengxiang Ge & Tingdong Li

Reviewer 1

Comments and Suggestions for Authors:The authors reviewed the pathway of homology-directed repair (HDR) and how to control it in response to CRISPR-directed double-strand break. Since genome editing technology is simple in principle of operation, but the process and output are often complicated, researchers need to be careful about how to use it. In this review manuscript, authors described HDR process and key players in comparison with other cellular processes including NHEJ and SSA. The manuscript is well-described in terms of the issues associating with both HDR and Cas9. I have some minor comments as follows:

1.Line 361-364; the word "dn53P1" should be corrected to "dn53BP1".

Re:  We are sorry for this error, “dn53P1” has been revised to “dn53BP1” (line 366-367).

2.Line 90; "S/G1 phase" should be corrected to "S/G2 phase".

Re:  Thanks for your comment and it has been revised (line98). 

3.Supplementary table 1 should be placed in the main text if possible, because of its importance.

Re: Thanks for your advice. Table 1 has been placed in the main text. (line 467). 

Dear Editors and Reviewers:

Thanks for your professional handling and consideration of our manuscript entitled “Methods Favoring Homology-Directed Repair Choice in Response to CRISPR/Cas9 Induced-Double Strand Breaks” (ijms-900814). The comments and advice from the reviewers are very helpful for revising and improving our manuscript. According to the reviewers’ comments and your advice, we revised relevant parts in the manuscript and tracked changes on the modified version of ijms-900814.In addition, the language has been edited by Charlesworth Author Services before submission. A file with line-to-line response to Reviewers’ comments was attached. I wish that the revised manuscript is now suitable to be published in “International Journal of Molecular Sciences”.

Thanks, and best wishes.

Yours sincerely,

Shengxiang Ge & Tingdong Li

Reviewer 1

Comments and Suggestions for Authors:The authors reviewed the pathway of homology-directed repair (HDR) and how to control it in response to CRISPR-directed double-strand break. Since genome editing technology is simple in principle of operation, but the process and output are often complicated, researchers need to be careful about how to use it. In this review manuscript, authors described HDR process and key players in comparison with other cellular processes including NHEJ and SSA. The manuscript is well-described in terms of the issues associating with both HDR and Cas9. I have some minor comments as follows:

1.Line 361-364; the word "dn53P1" should be corrected to "dn53BP1".

Re:  We are sorry for this error, “dn53P1” has been revised to “dn53BP1” (line 366-367).

2.Line 90; "S/G1 phase" should be corrected to "S/G2 phase".

Re:  Thanks for your comment and it has been revised (line98). 

3.Supplementary table 1 should be placed in the main text if possible, because of its importance.

Re: Thanks for your advice. Table 1 has been placed in the main text. (line 467). 

Reviewer 2 Report

The presented review is well constructed, quite updated, written in a concise manner and has a logical, easy-to-follow layout.

Minor revisions/suggestions

Line 43: “The type II system of Streptococcus pyogenes (Sp), requiring only one cas gene, cas9, is the most widely used for genome engineering purposes”. This is rather confusing because Streptococcus pyogenes CRISPR-Cas system actually needs cas1, cas2 and cas9 to be functional, you might just say that type II SpCas9 is the most widely used for genome engineering purposes.

Line 90: “HDR occurs largely during the S/G1 phase”.  you may mean S/G2 phase.

Lines 125-126: “Aligned, compatible DNA termini can be directly ligated by the c-NHEJ factors, if not exist…”. Please make this sentence clear: Aligned, compatible DNA termini can be directly ligated by the c-NHEJ factors, in case DNA termini were incompatible

Figure 2: please change DNK-PKcs by DNA-PKcs

Lines 172-173: “CDK activity also promotes long-range resection through the phosphorylation of Exo1 and NBS1”. Though the role of NBS1 in resection initiation is clear in the text and figures, you have not previously cited its role in long-range resection. Thus, within this sentence, do you refer to NBS1 or to BLM-DNA2 phosphorylation?

Figures 3 and 4: Please notice: Limited resection/ Extensive resection

Lines 228-229: “The mechanisms that allow 53BP1 to be replaced with BRCA1, and why it is it active at this specific time, was not fully elaborate”. Please remake this sentence. The mechanisms that allow 53BP1 to be replaced by BRCA1, and why it is it active at this specific time, was have not been fully elaborated yet”.

Line 275: “SCR7 is a potent NHEJ inhibitor…”

Lines 280-281: “The authors also observed that co-injection of CRISPR-Cas9 components and SCR7 into zygotes improved the efficiency of HDR-mediated insertional mutagenesis…”. You might point out that embryonic stem cells do preferably use HDR, thus, herein the role of SCR7 as an HDR promoter might be questionable.

Line 326: Please notice the mistake “Although serval studies…”

Line 336: do you mean MRE11?

Line 355: Cas9-mediated

Line 394: “to control the expression of hCas9…”. Though Gutschner et al use this term in their manuscript, might you change hCas9 by hSpCas9 and use every time Cas9-geminin instead of hCas9-geminin (line 399) to avoid misunderstandings?

Line 408: “…simultaneously inhibited Ku80 expression and stimulated CDK1” You might add respectively in order to make the sentence clearer: simultaneously inhibited Ku80 expression and stimulated CDK1, respectively

Line 408: please add Ye et al cite and include it in the references section

Figure 6: Please check figure 6 legend, it is confusing, change in by to: Methods restricting the Cas9-induced DSB to HDR-active cell cycle phase(s)

Line 423: Reference 125 corresponds to Ma, M et al. You wrote his first name Ming in line 421.

Figure 7: Please change Botin by Biotin, Advin by Avidin (or write this abbreviation in figure legend). Please change HUH target sequence

Supplemental Table 1: title: “Select small molecules used in favouring…”.

In table: HDR increase

Author Response

Dear Editors and Reviewers:

Thanks for your professional handling and consideration of our manuscript entitled “Methods Favoring Homology-Directed Repair Choice in Response to CRISPR/Cas9 Induced-Double Strand Breaks” (ijms-900814). The comments and advice from the reviewers are very helpful for revising and improving our manuscript. According to the reviewers’ comments and your advice, we revised relevant parts in the manuscript and tracked changes on the modified version of ijms-900814.In addition, the language has been edited by Charlesworth Author Services before submission. A file with line-to-line response to Reviewers’ comments was attached. I wish that the revised manuscript is now suitable to be published in “International Journal of Molecular Sciences”.

Thanks, and best wishes.

Yours sincerely,

Shengxiang Ge & Tingdong Li

Reviewer 2

Comments and Suggestions for Authors:The presented review is well constructed, quite updated, written in a concise manner and has a logical, easy-to-follow layout.

  1. Line 43: “The type II system of Streptococcus pyogenes (Sp), requiring only one cas genecas9, is the most widely used for genome engineering purposes”. This is rather confusing because Streptococcus pyogenes CRISPR-Cas system actually needs cas1, cas2 and cas9 to be functional, you might just say that type II SpCas9 is the most widely used for genome engineering purposes.

Re:  Thanks for your comment. The text has been revised to “The type II SpCas9 is the most widely used for genome engineering purposes” (line 50-52).   

  1. Line 90: “HDR occurs largely during the S/G1 phase”.  you may mean S/G2 phase.

Re:  Thanks for your comment and it has been revised (line 98). 

  1. Lines 125-126: “Aligned, compatible DNA termini can be directly ligated by the c-NHEJ factors, if not exist…”. Please make this sentence clear: Aligned, compatible DNA termini can be directly ligated by the c-NHEJ factors, in case DNA termini were incompatible

Re:  Thanks for your suggestion and the sentence is indeed much clearer. The text has been revised to “Aligned, compatible DNA termini can be directly ligated by the c-NHEJ factors, in case DNA termini were incompatible, one of the main exonucleases, artemis, cooperates with polymerase lambda (λ) and mu (μ) to prepare blunt DNA ends for the next stage” (line 133-135). 

  1. Figure 2: please change DNK-PKcs by DNA-PKcs

Re:  We are sorry for this error in Figure 2, and “DNK-PKcs” has been revised to “DNA-PKcs”.

  1. Lines 172-173: “CDK activity also promotes long-range resection through the phosphorylation of Exo1 and NBS1”. Though the role of NBS1 in resection initiation is clear in the text and figures, you have not previously cited its role in long-range resection. Thus, within this sentence, do you refer to NBS1 or to BLM-DNA2 phosphorylation?

Re:Thanks for your comments. We are very sorry that it was not described clearly enough. Within this sentence (line:179-180), we refer to phosphorylation of Exo1 in promoting long-range resection (line 179-181).

  1. 6. Figures 3 and 4: Please notice: Limited resection/ Extensive resection

Re: Thanks for your comments. We are sorry for the spelling error and “resction” has been revised to “resection” (Figure 3 and 4).

  1. Lines 228-229: “The mechanisms that allow 53BP1 to be replaced with BRCA1, and why it is it active at this specific time, was not fully elaborate”. Please remake this sentence. The mechanisms that allow 53BP1 to be replaced by BRCA1, and why it is itactive at this specific time, washave not been fully elaborated yet”.

Re: We are sorry for this grammatical error and we have revised it to “The mechanisms that allow 53BP1 to be replaced with BRCA1, and why it is active at this specific time, have not been fully elaborated yet” (line 234-235). 

  1. Line 275: “SCR7 is apotent NHEJ inhibitor…”

Re: Thanks for your comments. We are sorry for this error, and “a” has been added (line 279). 

  1. Lines 280-281: “The authors also observed that co-injection of CRISPR-Cas9 components and SCR7 into zygotes improved the efficiency of HDR-mediated insertional mutagenesis…”. You might point out that embryonic stem cells do preferably use HDR, thus, herein the role of SCR7 as an HDR promoter might be questionable.

Re: Thank you for your suggestion.We have pointed out that embryonic stem cells do preferably use HDR, thus, herein the role of SCR7 as an HDR promoter might be questionable ,and the text is clearer and more organized after revision (line 286-287). 

  1. Line 326: Please notice the mistake “Although servalstudies…”

Re: Thanks for your comments. We are sorry for the spelling error and it has been revised (line 340).

  1. Line 336: do you mean MRE11?

Re: Thanks for your comments. We are sorry that the error was not found before submission and it has been revised (line 331).

  1. Line 355: Cas9-mediated

Re: Thanks for your comments, and it has been revised (line 360).

  1. Line 394: “to control the expression of hCas9…”. Though Gutschner et aluse this term in their manuscript, might you change hCas9 by hSpCas9 and use every time Cas9-geminin instead of hCas9-geminin (line 399) to avoid misunderstandings?

Re: Thanks for your comments. We have revised it according to your requirements and it is easier to understand after the revision(line 399-404).

  1. Line 408: “…simultaneously inhibited Ku80 expression and stimulated CDK1”You might add respectivelyin order to make the sentence clearer: simultaneously inhibited Ku80 expression and stimulated CDK1, respectively

Re: Thank you for your suggestion, and the sentence is clearer after revision (line 412). 

  1. 15. Line 408: please add Ye et alcite and include it in the referencessection

Re: Thanks for your comments. Ye et al cite have been added and been included in the references section (Ref:122).

  1. Figure 6Please check figure 6 legend, it is confusing, change in by to: Methods restricting the Cas9-induced DSB toHDR-active cell cycle phase(s)

Re: Thanks for your comments. The figure 6 legend has been revised to“Methods restricting the Cas9-induced DSB to HDR-active cell cycle phases”(Figure 6).

  1. Line 423: Reference 125 corresponds to Ma, M et al. You wrote his first name Ming in line 421.

Re: Thanks for your comments. We are sorry for the error and it has been revised (line 426). 

  1. Figure 7: Please change Botinby Biotin, Advin by Avidin (or write this abbreviation in figure legend). Please change HUH target sequence

Re: Thanks for your comments, and these errors have been revised (Figure 7).

  1. Supplemental Table 1: title: “Select small molecules used in favouring…”.In table: HDR increase

Re: Thanks for your comments. We are sorry for these errors and they have been revised  (line 467).
